

# CALIPSO Lidar Calibration at 532 nm: Version 4 Daytime Algorithm

Brian J. Getzewich[1], Mark A. Vaughan[2], William H. Hunt[1,†], Melody A. Avery[2], Kathleen A. Powell[2], Jason L. Tackett[1], David M. Winker[2], Jayanta Kar[1], Kam-Pui Lee[1], Travis Toth[2,3]

[1]Science Systems and Applications Inc., Hampton, Virginia, USA
[2]NASA Langley Research Center, Hampton, Virginia, USA
[3]University of North Dakota, Grand Forks, North Dakota, USA
[†]deceased

*Correspondence to*: Brian J. Getzewich (brian.j.getzewich@nasa.gov)

**Abstract.** The Cloud Aerosol Lidar and Infrared Pathfinder Satellite Observations (CALIPSO) mission released version 4.00 of their lidar level 1 data set in April of 2014, and subsequently updated this to version 4.10 in November of 2016. The primary difference in the newly released version 4 (V4) data is a suite of updated calibration coefficients calculated using substantially revised calibration algorithms. This paper describes the revisions to the V4 daytime calibration procedure for the 532 nm parallel channel. As in earlier releases, the V4 daytime calibration coefficients are derived by scaling the raw daytime signals to the calibrated nighttime signals acquired within a calibration transfer region, and thus the new V4 daytime calibration benefits from improvements made to the V4 532 nm nighttime calibration. The V4 calibration transfer region has been moved upward from the upper troposphere to the more stable lower stratosphere. The identification of clear-air columns by an iterative thresholding scheme, crucial to selecting the observation regions used for calibration, now uses uncalibrated 1064 nm data rather than recursively using the calibrated 532 nm data as was done in version 3 (V3). A detailed account of the rationale and methodology for this new calibration approach is provided, along with results demonstrating the improvement of this calibration over the previous version. Extensive validation data acquired by NASA's airborne high spectral resolution lidar (HSRL) shows that during the daytime the average difference between collocated CALIPSO and HSRL measurements of 532 nm attenuated backscatter coefficients is reduced from $3.3\% \pm 3.1\%$ in V3 to $1.0\% \pm 3.5\%$ in V4.

## 1 Introduction

The Cloud-Aerosol Lidar with Orthogonal Polarization (CALIOP), on-board the Cloud-Aerosol Lidar and Infrared Pathfinder Satellite Observations (CALIPSO) satellite, has been providing a near continuous record of high-resolution vertical profiles of clouds and aerosols since the summer of 2006. Launched 28 April 2006, CALIPSO is an integral part of the NASA's Afternoon (A-Train) constellation, working in tandem with other Earth observing satellites to probe the nature and influence of clouds and aerosols on the global climate system (Winker et al., 2010). CALIOP is a dual wavelength, polarization-sensitive elastic backscatter lidar powered by a Nd:YAG diode-pumped laser that makes range-resolved measurements of the total backscatter intensity at 1064 nm and the 532 nm backscatter intensities in planes oriented parallel and perpendicular to the polarization plane of the transmitted laser beam (Hunt





et al., 2009). Among the first of its kind, and having delivered the longest duration of continuous on-orbit operations, CALIOP provides unique insights into the vertical distribution, morphology and variability of clouds and aerosols (Chand et al., 2009; Vernier et al., 2011; Forbes and Ahlgrimm, 2014; Tan et al., 2016; Stephens et al., 2018).

Measured CALIOP signals are translated into meaningful atmospheric observations by proper calibration of the three
receiver channels. Due to differences in signal-to-noise ratios (SNR) and intra-orbit thermal stability, CALIOP uses different techniques to calibrate the 532 nm daytime and nighttime measurements. The calibration procedure for the nighttime 532 nm parallel signals, which is the basis for all other calibrations, uses a high-altitude molecular normalization technique (Russell et al., 1979; Powell et al., 2009; Kar et al., 2018), in which calibration coefficients are determined by taking the ratio of the measured signal to the expected signal computed using an atmospheric model.
This approach assumes that all constituents of the nighttime normalization region (i.e., including aerosol loading) can be accurately modeled or characterized. The same technique cannot be used during daytime, however, because the SNR is substantially lower due to the influence of the reflected solar background radiation. This is rectified by scaling the daytime to the nighttime calibration by using clear air attenuated scattering ratios, defined as the ratio between the measured attenuated backscatter and modeled molecular signal. These are measured and accumulated over identical
altitude ranges and latitude bands during both daytime and nighttime.  The fundamental assumption for the 532 nm daytime calibration procedure is that a persistent "calibration transfer region" can be identified where the aerosol loading remains diurnally invariant over relatively short periods of time (e.g., 7–10 days).

Calibration algorithms used in the version 3 (V3) series of L1 data products (Vaughan et al., 2017), released beginning in June 2009 are described in Hostetler et al., 2005, Powell et al., 2009, Powell et al., 2010, and Vaughan et al., 2010.
Over the intervening years since the release of V3, several shortcomings have been identified in the 532 nm daytime calibration algorithm. First, the altitude of the V3 calibration transfer region was too low, and hence the assumed diurnal invariance for the 532 nm daytime calibration was often not satisfied. Frequent cloudiness at tropical latitudes also limited the number of clear-air samples available at this altitude range. Second, identifying the cloud-free data segments needed to calculate the V3 daytime calibration coefficients was accomplished by repeatedly generating a
subset of the lidar level 2 (L2) products, and this interdependency prohibited the independent calculation of the 532 nm daytime calibration coefficients. Finally, the calculation of the calibration uncertainty estimates in V3 failed to accurately include all error sources associated with the approach.

To account for these identified weaknesses in the V3 algorithm, the CALIPSO project completely redesigned the calibration architecture for the version 4 (V4) L1 data products.  The 532 nm nighttime calibration has been updated
to accommodate a change in the molecular normalization region in the stratosphere from 30-34 km to 36-39 km (Kar et al., 2018). This change was based on a better understanding of the vertical distribution of stratospheric aerosols (Vernier et al., 2009). As for the 532 nm daytime calibration, which is the focus of this paper, the technique still relies on matching daytime and nighttime clear air scattering ratios, but with several crucial modifications that address the problems identified above. An overview of the V3 calibration procedure will be provided in Sect. 2, followed by a
detailed summary of the new V4 calibration in Sect. 3. Section 3 fully describes the updated assumptions and new techniques used in V4. Section 4 compares the V4 calibration against internally established scientific metrics used to



assess the performance of the algorithm. Section 4 also updates a previous comparison between CALIOP V3 backscatter coefficients and extensive collocated measurements from the NASA Langley Research Center (LaRC) airborne high spectral resolution lidar (HSRL) (Rogers et al., 2011). Some concluding remarks are given in Sect. 5.

**2 Version 3 532 nm daytime calibration**

The V3 532 nm daytime calibration procedures transfer the 532 nm parallel channel nighttime calibration to the daytime measurements (Powell et al., 2010). Calibrating the daytime signals relative to the nighttime measurements is done for two reasons: (a) low SNR during the daytime prevents calibration using the high-altitude molecular normalization technique, and (b) thermally-induced changes in the alignment of the laser transmitter with respect to the receiver produce substantial changes in the daytime calibration over the course of each daytime orbit segment.

Transferring calibration from nighttime to daytime was accomplished in V3 by using latitudinally varying clear-air attenuated scattering ratios, accumulated for both day and night orbital segments, to derive correction factors associated with the along-track misalignments that occur during the daytime. For any daytime granule, the 532 nm daytime calibration coefficients were then computed as the product of these correction factors and the mean 532 nm calibration coefficient from the previous nighttime granule (Powell et al., 2010).

The attenuated scattering ratios, $R'$, are defined as the ratio between the measured total attenuated backscatter coefficients, $\beta'_{measured}$, and a profile of modeled molecular attenuated backscatter coefficients, $\beta'_{model}$, derived from modeled profiles of temperature and pressure (Powell et al., 2009), as given by

$$R'(z) = \frac{\beta'_{measured}(z)}{\beta'_{model}(z)} = \frac{X_{total}(z)/\tilde{C}}{\beta'_{model}(z)T^2_{model,m}(z)T^2_{model,O_3}(z)} . \qquad (1)$$

In this expression the subscripts m and $O_3$ represent, respectively, contributions from molecular and ozone scattering
and attenuation. $\tilde{C}$ is the estimated 532 nm total calibration coefficient, $T^2$ is the two-way transmittance between the lidar and altitude z, and $X_{total}$ is the range corrected, gain and energy normalized total backscatter signal at 532 nm (i.e., the sum of the parallel and perpendicular components), such that

$$X_{total}(z) = \left(\frac{r(z)^2}{E}\right)\left(\frac{P_{||}(z)}{G_{||}} + \frac{P_{\perp}(z)}{G_{\perp} \, PGR}\right), \qquad (2)$$

where $r(z)$ is the range from the lidar to altitude z, E is the laser pulse energy. $P_{||}(z)$ and $P_{\perp}(z)$ are, respectively, the
532 nm signals measured in the parallel and perpendicular channels, $G_{||}$ and $G_{\perp}$ are the electronic gains of the respective receiver channels, and PGR is the polarization gain ratio; i.e., "the ratio of $P_{||}(z)$ to $P_{\perp}(z)$ when both channels are illuminated by the same light levels" (Hunt et al., 2009; Powell et al., 2009).

The nighttime and daytime clear-air attenuated scattering ratios in V3 are calculated for "frames" of data within the calibration transfer region. Each frame extends for 200 km along-track at an altitude of 8 to 12 km. The 8 to 12 km
altitude range was chosen to be high enough to avoid substantial diurnal variation of the aerosol loading in the lower





troposphere and low enough to provide increased SNR, thus minimizing the influences of solar background radiation on the daytime signal. The clear-air attenuated scattering ratios in the V3 8-12 km calibration transfer region were assumed to be diurnally invariant. Based on this assumption, initial estimates of the mean attenuated scattering ratios, $\overline{R}'_{day,initial}$, are calculated for each daytime frame using the mean of the 532 nm calibration coefficients, $\tilde{C}_{night}$, computed during the previous nighttime granule; i.e.,

$$\overline{R}'_{day,initial} = \left\langle \frac{X_{total,day}(z_j,p_k)}{\tilde{C}_{night}\beta'_{model,day}(z_j,p_k)} \right\rangle. \tag{3}$$

Here the angle brackets indicate averaging over all altitudes $z_j$ between 8 km and 12 km for all profiles $p_k$ lying within the daytime clear-air calibration transfer region. For each nighttime clear-air region, the mean clear-air attenuated scattering ratios, $\overline{R}'_{night}$, are calculated using the same formula as in Eq. 2, except using the nighttime signals, gain settings, and calibration coefficients.

Initial correction factor estimates, $W_{initial}$, are then derived by using the mean attenuated scattering ratios residing in corresponding latitudes of the day-night calibration transfer regions of the orbits, as seen in Eq. (4):

$$W_{initial} = \frac{R'_{day,initial}}{R'_{night}} \tag{4}$$

The 532 nm daytime calibration procedure generates daily estimates of correction factors built from a moving averaged window of the day and night clear-air attenuated scattering ratios accumulated over the previous seven days. There is a minimum of four days required in case of instrument downtime or the inability to downlink data. Looking backwards, and not including current scattering contributions for the orbit that is being calibrated, is required because the L2 5 km cloud and aerosol layer data products are needed to identify the clear air regions. These daily correction factors are smoothed onto a 1° latitude window and reported as a function of elapsed seconds, $W(t)$, from the start of some reference orbit that reflects the general orbital configuration for that day (i.e., time will differ from latitude based on time of year). Additional filtering and smoothing are applied to mitigate outliers. In particular, a minimum nighttime scattering ratio of 1.03 is used to compensate for diurnal differences in aerosol loading in the troposphere. Selection of this offset was based on observational analysis during development of the V3 algorithm, where it was noted that zonal distributions of attenuated scattering ratios in the calibration transfer regions fell below 1.0 in the tropics. This dip in the nighttime scattering ratios is attributed to signal attenuation by undetected cloud and aerosol layers in the upper troposphere (e.g., Vernier et al., 2009).

To construct valid sets of correction factors, the data frames used in the V3 daytime calibration procedure must consist entirely of "clear air" from the top of the lidar return at 40 km down to the base of the V3 calibration transfer region at 8 km. Note that, in this context, "clear air" does not imply a pristine, aerosol-free molecular atmosphere. Instead, we impose the requirement that no layers are detected within this region by the CALIOP multi-resolution layer detection algorithm (Vaughan et al., 2009). Any undetected residual aerosol is assumed to be diurnally invariant. For the calibrated nighttime data, the required clear air frames can be identified directly from the automatically generated L2 data products. But because the layer detection algorithm requires calibrated L1 data for its operation, determining





daytime clear air frames requires an intermediate operation, wherein the layer detection module is embedded in an iterated optimization loop initiated using a coarse first approximation to the daytime calibration scale factors. These scale factors are subsequently refined through successive iterations. At each step, the required regions of "clear air" are identified by applying the layer detection procedure to the current estimate of the daytime attenuated scattering

ratios, which are derived by applying the most recent iteration of the daytime calibration scale factors. The final correction factor curves are stored in internal ancillary lookup tables. The 532 nm daytime calibration coefficients are derived by multiplying these time-varying correction factors by the mean of the previous granule's 532 nm nighttime calibration coefficient, as shown in Eq. (5), where t represents granule-elapsed time; i.e.,

$$C_{532,day}(t) = W(t)\tilde{C}_{night}. \tag{5}$$

Figure 1 shows the application of Eq. (5) used to build the 532 nm calibration coefficients for a daytime orbit on 18 December 2016. No effort was made to anchor the V3 532 nm daytime calibration with the adjoining V3 nighttime calibrations, which causes abrupt discontinuities in the dayside calibration trend at both terminators of the daytime orbit.

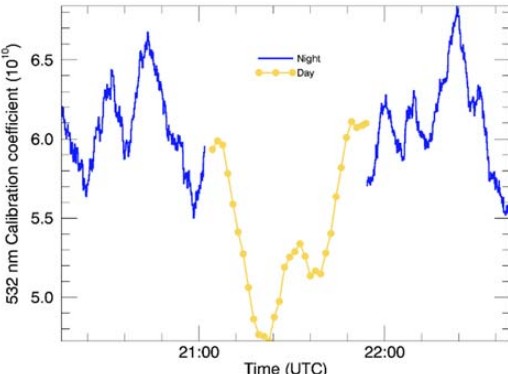

**Figure 1: V3.40 532nm calibration coefficients between successive CALIPSO night-day-night orbits on December 18, 2016 from 20:15:31 to 22:40:41 UTC.**

### 3 Version 4 532 nm daytime calibration

The fundamental assumption underlying the V4 532 nm daytime calibration scheme is the same one invoked in V3; i.e., the mean attenuated scattering ratios do not vary significantly during a diurnal cycle within a defined region in

the atmosphere, and hence the mean uncalibrated daytime attenuated scattering ratios can be scaled to match the mean nighttime attenuated scattering ratios in this calibration transfer region. Here we list the major changes between the V3 and V4 daytime calibration algorithms. First, two new signal adjustments have been incorporated into the L1 processing, and these have a direct impact on the subsequently derived calibration coefficients. Second, the selection of the calibration transfer region has been changed so that a 400 K isotherm in the lower stratosphere now defines the

bottom of the vertical calibration range, replacing the fixed altitude cross-tropopause region used in V3. Third, a


newly developed multi-granule averaging scheme compensates for the reduced SNR incurred by moving the calibration transfer region upwards. To further boost the SNR, rather than using the total signal (i.e., parallel + perpendicular, as in Eq. (2)), only the parallel channel is considered. Fourth, the V4 procedure uses a modified version of the L2 layer detection algorithm to search the uncalibrated 1064 nm channel backscatter coefficients for clear air

regions, instead of using the calibrated 532 nm attenuated scattering ratios, as was done in V3. This eliminates the need for a multi-pass architecture, and has two major benefits. The revised V4 calculations are more transparent, making it easier for external data users to replicate and/or validate the calibration coefficients and uncertainties reported in the V4 L1 data products. Also, by eliminating the recursive process, the V4 scheme significantly reduces the number of processing steps required to generate the resultant data product. Fifth and finally, the V4 532 nm

daytime calibration coefficients uncertainties are computed directly from the nighttime calibration coefficient uncertainties, using the calibrated 532 nm nighttime and uncalibrated 532 nm daytime attenuated scattering ratios. Increased accuracy in computing the 532 nm daytime calibration coefficient uncertainties for V4 was also important, as a better understanding of the key contributors to the overall error was crucial in driving the averaging decisions used to calibrate all three channels. Each of these five algorithm updates is discussed in detail in the subsections below.

**3.1 Signal adjustments**

The V4 calibration procedure applies two new corrections to the daytime signal prior to the calibration: adjustment of the baseline slope correction and an updated day-to-night gain ratio. During daytime operations when the instrument is on but the laser is not actively firing, the detector response to background signals from scattered sunlight varies slightly with altitude. This baseline slope varies with the magnitude of the background light, and the response was

modelled using laboratory measurements prior to launch. Post-launch, the baseline slope has been characterized on-orbit during periods of extended background measurements, with the receiver on and the laser turned off. Comparisons between pre-launch results and several years of collected on-orbit extended background measurements formed the basis for the baseline slope correction applied in V4. For V3, it was expected that the impact of the baseline slope on the daytime calibration would be negligible, because the small slope errors would be mitigated by the relatively large

signal in the 8-12 km calibration transfer region, and the biggest impact would occur at high altitudes where the signal is much lower relative to the baseline perturbations. The accuracy of the baseline slope correction becomes more important in V4 because the calibration transfer region is located higher in the atmosphere, as discussed in more detail in Sect. 3.2, where the magnitude of the molecular signal is substantially lower.

Figure 2 shows a 7-day average (13–19 December 2011) of the daytime correction factors derived using the V3

algorithm as a function of orbital elapsed time for both the 8-12 km (V3 calibration transfer region) and an elevated 18-22 km region (an approximate V4 calibration transfer region for this time period). Figure 2(a) shows that there is a difference of up to 8% in the correction factors between these two altitude levels, particularly in mid-latitudes where bright clouds generate higher background signals. This difference as a function of altitude should not occur. By applying the V4 baseline slope correction for this case in Fig. 2(b), the correction factors, though slightly reduced, are

now more similar. Although applying the slope correction causes an overall reduction in the apparent signal with





increased altitude, the corrected signal more accurately corresponds to the atmospheric signal and eliminates systematic artifacts in the daytime calibration coefficients.

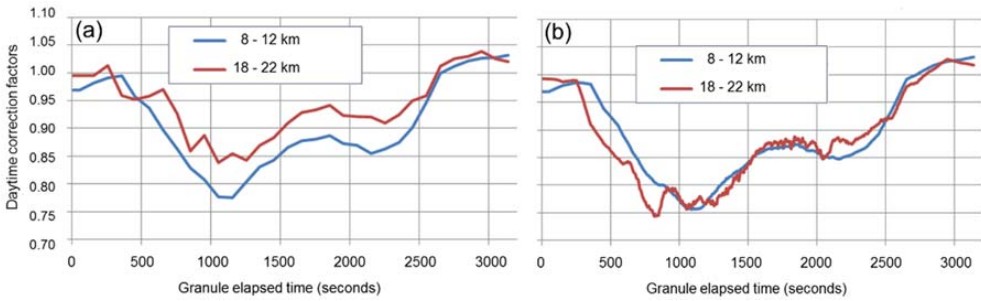

**Figure 2: The 532 nm daytime correction factor for December 13-19, 2011 based on the V3 L1 algorithm. The correction**
**factor is computed for both the V3 calibration transfer region (8-12km) and an elevated transfer region (18-22km) without**
**(a) and with (b) the baseline slope correction applied to the signal.**

To prevent saturation of the digitizers by large daytime noise excursions, a fixed reduction in the detector gains is applied to all three channels during daytime operations. To compensate for these gain changes, the CALIOP calibration routine applies fixed day-to-night gain ratios to the daytime measurements. On-orbit performance metrics

and routine built-in test system (BITS) measurements (Hunt et al., 2009) suggested that the 532 nm perpendicular and parallel day-to-night gain ratios needed to be increased by 0.65 % and 3.3 %, respectively. Though large, the 532 nm parallel adjustment has essentially no impact on the calibrated 532 nm daytime attenuated backscatter coefficients, because the gain increase is absorbed as a multiplicative factor into the calculated calibration coefficients. Similarly, because the V4 daytime calibration only uses the signals from the parallel channel, changes to the 532 nm

perpendicular day-night gain ratios have no impact on the derived calibration, though they will ultimately yield a small increase in the 532 nm perpendicular and total attenuated backscatter coefficients reported in the L1 data products.

**3.2 Revised calibration transfer region**

The selection of the calibration transfer region is subject to two competing interests. Diurnal variation in background aerosol should be minimized, which argues for a higher altitude since, to first order, aerosol concentrations tend to

decrease with height. However, absent any aerosol loading CALIOP's SNR also decreases with height, and obtaining an accurate calibration requires sufficient signal to overcome the daytime background noise due to sunlight. The V3 algorithm approach maximized SNR, as previously discussed, by choosing a calibration transfer region with a fixed base of 8 km and a constant depth of 4 km (12 km top). However, this altitude domain occurs in the tropical troposphere where the CALIOP signal is frequently attenuated due to persistent cloud cover, and therefore has a reduced number

of clear-air samples. There is also the possibility of potential contamination by clouds and aerosols not identified by the L2 feature detection technique used to isolate clear air. In the extra-tropics, 8–12 km altitude range straddles the tropopause, where there is additional background aerosol variability caused by fluctuations in the tropospheric jet locations (Gettleman and Wang, 2005; Manney and Hegglin, 2018). By elevating the calibration transfer region from the near tropopause into the lower stratosphere, the V4 approach attempts to improve the fidelity of the clear-air





attenuated scattering ratios by substantially reducing the possibility of any diurnal variability of background aerosol. Relocating to the lower stratosphere also minimizes the need for a robust feature detection algorithm to identify clear-air, as by definition this more stable region contains fewer cloud and aerosol layers than are found in the troposphere. The trade off, as already noted, is a reduction in SNR that dictates more sampling, which will be discussed in more

detail in Sect. 3.3.

Rather than using a globally fixed geometric altitude range, the base of the V4 calibration transfer region is located above the 400 K isentropic surface, with a thickness of 4 km, thereby reflecting latitudinal differences in the height of the lowermost stratosphere. Using an isentropic surface to identify the stratosphere is beneficial because (1) it accounts for latitudinal and seasonal changes in geopotential heights, and (2) potential temperature (θ) is likely to be a more

accurate field in the reanalysis data than are tropopause heights. This is because θ is constrained by basic atmospheric physics, while tropopause height estimates rely on assimilated global temperature measurements that have limited accuracy and a coarse vertical resolution (Reichler et al., 2003). The authors conducted an analysis of zonal potential temperature surfaces for multiple months in 2010, and concluded that a constant surface of 400 K reliably identifies the base altitude of the lower stratosphere with sufficient accuracy for use in the V4 CALIOP daytime calibration

procedure. The work product from this study is a comprehensive set of lookup tables derived from 5 years of V3 L2 5 km cloud profile data and the corresponding GEOS 5 FP-IT (Forward Processing for Instrument Teams) Version 5.91 meteorological data products distributed by NASA's Global Modeling and Assimilation Office (GMAO). These tables specify the geometric height field (km) corresponding to the 400 K isentropic surface, indexed by month and latitude, and are now used operationally by the V4 calibration algorithm to set the base of the calibration transfer

regions. The increased stability of this region (Hoskins, 1991) should act to cap motions from the lower troposphere, with the exceptions of strong diabatically forced events (deep convection, orographic gravity waves, volcanic events, etc.).

Two additional safeguards are used to avoid possible contamination of the clear-air attenuated scattering ratios. First, to both guard against features intruding into the lower stratosphere, and because the algorithm uses a climatological

monthly mean 400K surface as the lower limit, , an additional altitude offset of 2 km is applied to further elevate the base of the calibration transfer region. Secondly, since the stratosphere is not entirely devoid of features, the algorithm employs a 1064 nm feature detection technique, as discussed in Sect. 3.4 to remove potential features.

At the time of the V4 algorithm development and deployment, GMAO provided an updated meteorological reanalysis product, MERRA-2 (Modern-Era Retrospective analysis for Research and Applications, Version 2) (Gelaro, 2017),

which includes Microwave Limb Sounder (MLS) temperatures and is a marked improvement over earlier GMAO-FPIT products. This new meteorological data was incorporated into the V4.10 L1 and L2 data products, but was not used to re-compute the 400 K altitudes used by the 532 nm daytime calibration algorithm to set the calibration transfer region base altitude.



### 3.3 Mitigation of reduced SNR

Because the dominant source of noise during daytime operations is the solar background signal, daytime SNR scales approximately linearly with signal strength (Hunt et al., 2009). Moving the calibration region upward from a midpoint of 10 km in V3 to a nominal midpoint of ~20 km in V4 lowers the magnitude of the molecular attenuated backscatter

coefficients in the calibration transfer region, and hence the SNR, by a factor of ~5. To compensate for this significant reduction in the SNR at the higher calibration altitudes, the number of frames averaged in V4 must increase when compared with V3. Furthermore, because daytime SNR scales as the square root of the number of frames averaged, maintaining the same calibration SNR in V4 that was achieved in V3 requires the V4 procedure to accumulate ~25 times more frames than were used in V3. Given that each frame extends for 200 km along-track, this increase in

sample size is not something that can be accomplished within a single granule. Accurately characterizing the magnitude and rate of change of the daytime calibration coefficients within the V4 algorithm therefore requires averaging across-track over multiple consecutive daytime granules. Applying standard propagation of errors techniques to the daytime calibration equations shows that an averaging period of 105 consecutive orbits (i.e. over 7 days), centered on the orbit to be calibrated, should be sufficient to derive calibration coefficients with acceptably low

random uncertainties.

Continuous operation of the V4 daytime calibration procedure is predicated on maintaining the instrument in "steady state" conditions. There have been, of course, numerous cases where these steady state conditions have been interrupted. In these cases, the data acquired before and after the change in instrument state is potentially quite different, and hence the calibration procedure needs to be restarted at the temporal boundary of the change. The most

common changes occur when there are gaps in the data that exceed 24 hours. These are chiefly due to data dropouts, failed downlinks, or commanding of satellite to protect the hardware from solar storms or other anomalies. State changes also occur when routine on-orbit maintenance is performed. These tasks include re-alignment of the laser, etalon scans, or BITS (Sect. 3.2), and are followed by a reboot of the calibration procedure. This reboot introduces a hard-boundary, in which the calibration averaging windows stop at a defined time. Following a reboot, the calibration

coefficients are observed to remain quite stable. However, as expected, the calibration uncertainties increase, reflecting the lower numbers of samples used in the averaging.

The use of multi-orbit averaging also helps suppress the influence of the unusually large noise excursions that can occur when the satellite passes through the South Atlantic Anomaly (SAA), an area on the globe from roughly 90°W to 30°E in longitude and 0° to 45°S in latitude in which there is a greater influx of energetic particles than over the

rest of the globe. In general during the daytime, this increased radiation is largely indistinguishable from the solar background noise, but it has a greater impact on the nighttime calibration. Averaging data limited to non-SAA orbits at night provides a more stable clear-air scattering ratio for referencing the corresponding daytime measurement.





### 3.4 Identifying clear air attenuated scattering ratios

Because only clear-air regions are used for the 532 nm daytime calibration, frames are excluded where features are detected. In the CALIOP L2 processing, cloud and aerosol layers are detected using an iterated multi-resolution averaging scheme, in which the measured 532 nm attenuated scattering ratios are compared to dynamically

constructed, altitude-dependent threshold arrays (Vaughan et al., 2009). Large positive excursions in scattering relative to the computed thresholds are identified as features, and the spatial and optical properties of these features are subsequently used to discriminate clouds from aerosols (Liu et al., 2009; Liu et al., 2018) and then determine either cloud thermodynamic phase (Hu et al., 2009) or aerosol species (Omar et al., 2009; Kim et al., 2018). This same layer detection scheme was used in the V3 532 nm daytime calibration procedure to identify the presence of clouds and

aerosols on a per-frame basis within the daytime calibration transfer region (Powell et al. 2010).

The V4 daytime calibration scheme takes a different approach. Layers are still detected using the same profile scanning engine that drives the L2 processing. However, instead of recursively searching the calibrated 532 nm attenuated scattering ratios, layers are detected using the uncalibrated 1064 nm signals. In conducting the search, the molecular backscatter contribution to the total 1064 nm signal is assumed negligible, and the expected molecular

signal is set to zero. This assumption is reasonable because the large amount of dark noise from the avalanche photodiode (Hunt et al., 2009) is much larger than any molecular contribution in the stratosphere. Additionally, the search for layers is instead carried out at a single horizontal resolution of 200 km rather than using the iterated multi-resolution averaging scheme.

The 1064 nm threshold arrays for feature detection are constructed as follows. First, the measured background

variation, MBV (see Vaughan et al., 2005 for the approach used for the 532 nm detection method) in the averaged profile is computed using

$$\text{MBV} = \frac{\sqrt{\sum_{i=0}^{i=N} \text{RMS}_{1064}(i)^2}}{N}, \tag{6}$$

where N represents the number of profiles averaged and $\text{RMS}_{1064}$ is the root-mean-square of the baseline signal measured on-board the satellite for each laser pulse at 15 m vertical resolution and subsequently recorded in the L1

data products. The layer detection threshold, T(z), is then computed as a function of the on-board averaging using

$$T(z) = C_0 \, \text{MBV} \, F(r), \tag{7}$$

where F(r) accounts for apparent changes in noise magnitudes introduced by CALIOP's on-board data averaging scheme (Vaughan et al., 2009). F(r) is constant within each averaging region, but varies from region to region. Between 30.1 km and 20.2 km, F(r) = 1.2909944; between 20.2 km and 8.2 km, F(r)= 2.236068. $C_0$ is a scaling

constant that adjusts the magnitude of T(z) relative to MBV. For the V3 daytime calibration procedure, $C_0$ at 532 nm is set to 1.5; however for the 1064 nm uncalibrated signal used in V4, $C_0$ is set to 3.0. The value of $C_0$ in V4 is determined by matching the detection results achieved using the 532 nm scheme in V3; i.e., by lowering the calibration base altitude into the troposphere for multiple orbits and then comparing the frequency and altitudes of 1064 nm



feature detections against the L2 532 nm feature detections. The V4 layer detection scheme only needs to determine that a feature is present somewhere within or above the calibration transfer region. Features are identified whenever the 1064 nm signal excursions extend continuously above T(z) for 1 km or more. Layer identification (e.g., type and vertical extent) is not needed, since contamination of any type is grounds for excluding a region from use in the

remainder of the calibration procedure.

A detailed example of this technique is shown in Fig. 3, where the new 1064 nm feature detection algorithm evaluates a fairly typical blended cloud/aerosol scene for a nighttime granule. The 532 nm total attenuated backscatter is shown in Fig. 3(a) and the vertical feature mask (VFM) is shown in Fig. 3(b). Superimposed on Fig. 3(b) are the frames corresponding to the two calibration transfer regions: V3, indicated by red boxes between 8 and 12 km, and V4,

indicated using green boxes which track 2 km above the 400K isentropic surface. Clearly, the V4 calibration transfer region is well above all features detected within the vertical feature mask.

To further demonstrate the effectiveness of the new detection technique, the calibration base altitude was lowered to 8 km, matching the base of the V3 calibration transfer region. The top altitude at which the 1064 nm technique detected a feature is highlighted by a solid white line in Fig. 3(b). The detection follows the top of the cloud features, identifying

a deep convective cloud at 18° N and a transparent cirrus cloud from 15° N to 9.5° S. The lower panels of Fig. 3 compare the detection thresholds, T(z) (in red), to 1064 nm uncalibrated backscatter signals (in black) for three distinct features identified by the V3 L2 feature detection algorithm: clear-air (Fig. 3(c)), convective clouds (Fig. 3(d)), and cirrus clouds (Fig. 3(e)). The detection thresholds in these profiles are well selected to capture the clouds also detected by the V3 L2 algorithm in Fig. 3(d) and Fig. 3(e). Meanwhile, the requirement that the signal exceeds T(z) for 1 km

or more consecutive range bins prohibits false feature detections in the clear-air region (Fig. 3(c)). A more comprehensive validation of this 1064 nm feature detection technique is discussed in Sect. 4.3.





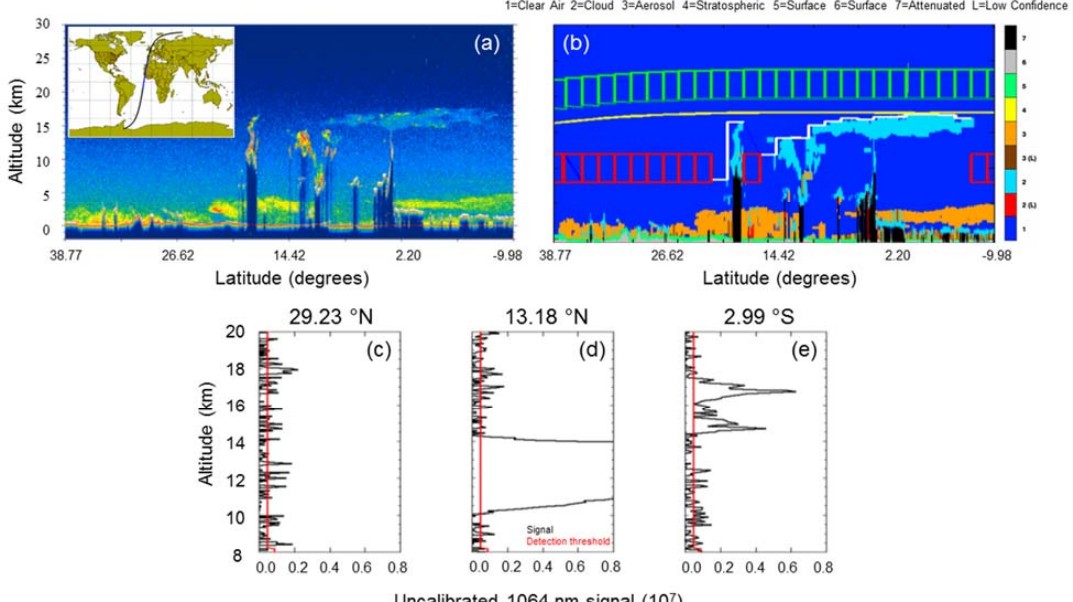

**Figure 3: (a) 532nm total attenuated backscatter and (b) vertical feature mask derived from the V3.01 Lidar Level 1 and Level 2 data products for a nighttime orbital segment in October 13, 2010. For (b) the top of the features detected by the 1064 nm technique are identified by a solid white line, V3 calibration transfer regions are identified as red boxes, V4 calibration target regions are identified as green boxes, and the potential temperatures surface of 400K is a solid yellow line. Profiles of uncalibrated 1064 nm signal with the applied detection threshold are shown in (c) – (e) for differing scenes contained in the orbit; clear, convective and cirrus respectively.**

### 3.5 Derivation of attenuated scattering ratio and scattering ratio uncertainty

The mathematical approach used to derive the 532 nm mean attenuated scatter ratios is fundamentally the same between V3 and V4 (Powell et al., 2010). Attenuated scattering ratios and uncertainty are averaged for frames of data within the calibration transfer region. A 'frame' is defined as 200 km along-track (index j) and 4 km vertical (index i, from base to top of the calibration transfer region in altitude) segments of data. Along-track, the 200 km resolution translates to 600 single shot (1/3 km native resolution) profiles. Those 1/3 km profiles that are considered invalid due to any errors or anomalies in the signal are removed. Frames are excluded if they contain features identified by the 1064 nm feature detection algorithm summarized in Sect. 3.4.

Expanding Eq. (1), the nighttime attenuated scattering ratios $\overline{R}'$ averaged for each frame are defined by

$$\overline{R}' = \overline{\sum_{J=0} \sum_{I=Base}^{Top} \frac{\beta'(I,J)_{532,\parallel,measured}}{\beta'(I,J)_{532,\parallel,model}}}. \tag{8}$$

The nighttime attenuated scattering ratio uncertainties, $\Delta\overline{R}'$, averaged for each frame, is given by

$$\Delta\overline{R}' = \overline{R}' \sqrt{\left(\sum_{J=0}\sum_{I=Base}^{Top}\left(\frac{\Delta X'(I,J)_{532,\parallel,measured}}{X'(I,J)_{532,\parallel,measured}}\right)\right)^2 + \left(\sum_{J=0}\frac{\Delta C(J)_{532,\parallel,night}}{C(J)_{532,\parallel,night}}\right)^2 + \left(\sum_{J=0}\sum_{I=Base}^{Top}\left(\frac{\Delta\beta'(I,J)_{532,\parallel,model}}{\beta'(I,J)_{532,\parallel,model}}\right)\right)^2} \tag{9}$$





This error quantifies the uncertainty associated with the measured 532 nm parallel uncalibrated attenuated backscatter, the 532 nm molecular attenuated backscatter, and the 532 nm nighttime parallel channel calibration coefficient, $C_{532,\|,night}$. Given that the night-time calibration coefficient is reported only on a per profile basis within the data frame, its uncertainty contribution has to be accounted for differently than the backscatter components, which are averaged both horizontally and vertically. The derivation and scale of the 532 nm night-time calibration error term is described in more detail by Powell et al. (2009). A more detailed derivation of the 532nm attenuated backscatter uncertainties can be found in Hostetler et al. (2006) and Liu et al. (2006).

The daytime averaged uncalibrated scattering ratios, $\overline{Q}'$, and uncertainties, $\Delta\overline{Q}'$, are computed using

$$\overline{Q}' = \overline{\sum_{J=0}\sum_{I=Base}^{Top} \frac{X'(I,J)_{532,\|,measured}}{\beta'(I,J)_{532,\|,model}}} \tag{10}$$

and

$$\Delta\overline{Q}' = \overline{Q}'\sqrt{\left(\sum_{J=0}\sum_{I=Base}^{Top}\left(\frac{\Delta X'(I,J)_{532,\|,measured}}{X'(I,J)_{532,\|,measured}}\right)\right)^2 + \left(\sum_{J=0}\sum_{I=Base}^{Top}\left(\frac{\Delta\beta'(I,J)_{532,\|,model}}{\beta'(I,J)_{532,\|,model}}\right)\right)^2}. \tag{11}$$

Because $\overline{Q}'$ uses uncalibrated data, accounting for the contribution of the 532 nm nighttime calibration error is not required.

### 3.6 Derivation of calibration coefficient and calibration coefficient uncertainty

The 532 nm calibration coefficients and their uncertainties for any given daytime granule are computed on a fixed elapsed time grid that spans from 0 seconds (referenced to the start of the daytime granule) to 3200 seconds with a resolution of 100 seconds. These calibration coefficients are then interpolated, based on time, so that they can be applied to the 532 nm attenuated backscatter measurements at the 1/3 km native resolution.

Given the multi-day averaging needed to harvest the calibration data, as described in Sect. 3.3, time cannot be used, either elapsed or some other reference time, to aggregate day and night scattering ratios that span multiple orbits. The orbital transition point from day to night (i.e., the day-night terminator), by which CALIPSO designates granules as either daytime or nighttime, changes throughout the aggregation period. In order to properly account for this temporal drift, a reference latitude grid, independent of time, is built by mapping and interpolating the latitude of the daytime granule that the algorithm is deriving the calibration for onto the fixed elapsed time grid.

The parallel component of the 532 nm daytime calibration coefficient, $C_{532,\|,day}$, is derived for each granule elapsed time grid cell (index k) using

$$C_{532,\|,day}(k) = \frac{\langle \overline{Q}'_{532,\|}\rangle_k}{\langle \overline{R}'_{532,\|}\rangle_k} = \frac{\frac{1}{N}\sum_{n=0}^{N-1}\overline{Q}'_{532,\|,n}}{\frac{1}{M}\sum_{m=0}^{M-1}\overline{R}'_{532,\|,m}}. \tag{12}$$



N is the number of aggregated daytime scattering ratio frames and M is the number of night-time scattering ratio frames contained in each elapsed time grid cell. $C_{532,||,day}$ is simply the ratio between the mean of the 532 nm un-calibrated daytime scattering ratios and the mean of the 532 nm calibrated night-time scattering ratio.

The daytime calibration uncertainty estimate, $\Delta C_{532,||,day}$, has contributions from both the 532 nm daytime and nighttime scattering ratio errors, as follows:

$$\Delta C_{532,||,day}(k) = C_{532,||,day}(k) \sqrt{\left(\frac{\Delta \bar{Q}'_{532,||}}{\bar{Q}'_{532,||}}\right)^2_k + \left(\frac{\Delta \bar{R}'_{532,||}}{\bar{R}'_{532,||}}\right)^2_k}. \tag{13}$$

### 3.7 Accommodating missing data

Calibration coefficients derived from the day-to-night ratio of attenuated scattering ratios can only be calculated within those portions of an orbit in which both day and night observations are acquired. Because of the illumination patterns in the polar regions during the solstice seasons there are no matching daytime and nighttime samples near the poles in summer and winter. This seasonal high latitude lack of matching day and night data is accounted for by anchoring the ends of the neighboring daytime and nighttime calibration coefficient curves and interpolating between these end points. Where there are neither daytime and/or nighttime samples, the 532 nm daytime calibration coefficient and coefficient uncertainty curves are linearly interpolated as a function of orbital elapsed time anchored to the nearest neighboring 532 nm nighttime calibration. Figure 4 shows an example. In this orbit from July 2010, the day-to-night terminator occurs at ~60° N, and thus no corresponding nighttime measurements are available over the final ~1100 seconds of the daytime granule.

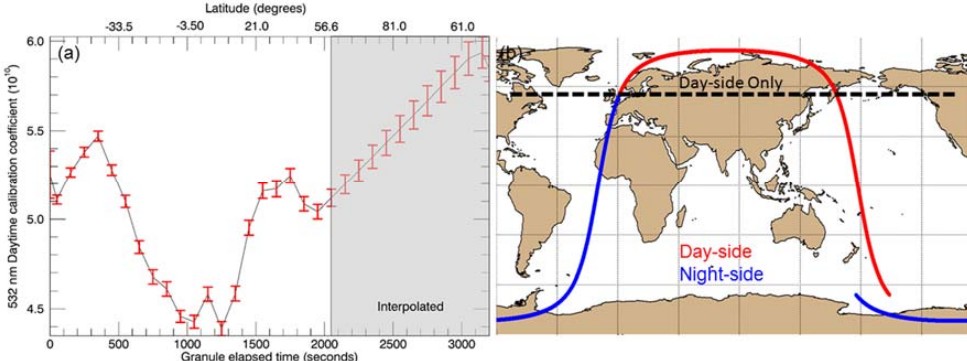

**Figure 4: (a) 532 nm daytime calibration and error for a July 2nd, 2010 daytime granule. The calibration and uncertainty for the high latitude segment (> 2050 seconds in the gray shaded region) are anchored and linearly interpolated to the next nighttime orbit. (b) The orbit track (red line) for which the calibration coefficients shown in panel (a) were derived, and the adjoining night-side orbit track (blue line). That portion of the orbit which is interpolated because of the lack of any night-side measurements is indicated by the black dashed black line.**





### 3.8 Calculating profiles of total attenuated backscatter coefficients

532 nm calibration coefficients for both daytime and nighttime measurements are computed using only the parallel component of the backscattered signal. The perpendicular channel measurements are calibrated relative to the parallel channel using the polarization gain ratio (PGR), which quantifies the relative gain between the two 532 nm detectors.

Highly accurate PGR values are measured directly using an onboard calibration procedure described in detail in Hostetler et al., 2005, Hunt et al., 2009, and Powell et al., 2009.  The calibration coefficients for the perpendicular channel are the product of the PGR and the parallel channel calibration coefficients; i.e.,

$$C_{\perp} = PGR \times C_{\parallel} \tag{14}$$

(Powell et al., 2009).  Given measured profiles of $P_{\parallel}(z)$ and $P_{\perp}(z)$, the profiles of 532 nm total attenuated backscatter

coefficients, $\beta'(z)$, reported in the CALIOP level 1 data product are derived using

$$\beta'(z) = \left( \frac{r(z)^2}{E} \right) \left( \frac{P_{\parallel}(z)}{G_{\parallel} C_{\parallel}} + \frac{P_{\perp}(z)}{G_{\perp} C_{\perp}} \right) \tag{15}$$

An identical procedure is followed for generating nighttime profiles of $\beta'(z)$ (Kar et al., 2018).

### 4 Verification and validation

### 4.1 Mission level performance

Performance of the 532 nm daytime calibration from 13 June2006 to 31 December 2016 is shown in Fig. 5. Figure 5(a) shows the 532 nm daytime calibration coefficient anomalies while Fig. 5(b) shows the 532 nm daytime calibration uncertainty anomalies, both of which are normalized to their respective time-series mean. The figure shows gaps in the data record occurring over the course of the mission, reasons for which are described in more detail in Sect. 3.3. From the start of the mission to 31 December 2016 there have been 138 distinct events that required calibration restarts,

with 71 of these due to planned maintenance of the lidar. The others were due to unscheduled events when either the instrument was commanded to SAFE/OFF (leading to a period when no data was collected) or when data downlink issues caused delays that exceeded 24 hours, and thus required a reboot of the calibration averaging, as described in Sect. 3.3. Also of note, the lidar switched from the primary laser to backup laser on 12 March 2009, resulting in a noticeable shift in the distributions of the calibration minimum between the two lasers.





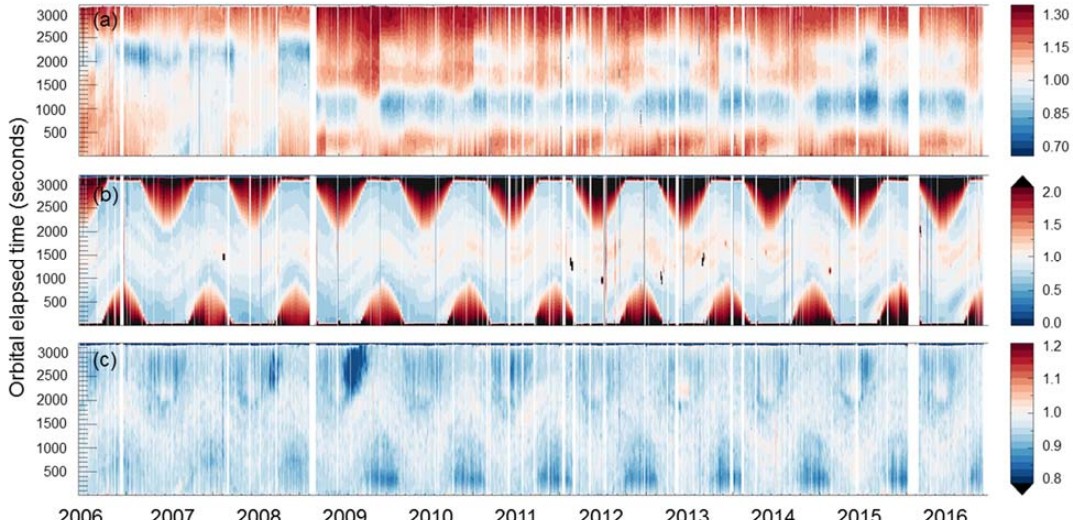

**Figure 5: Time series of (a) V4 532 nm daytime calibration coefficient anomalies, (b) calibration coefficient uncertainty anomalies, and (c) V4/V3 calibration ratio for June 13, 2006 to December 31, 2016 as functions of granule elapsed time. The calibration coefficients and uncertainties are extracted from the V4 and V3.x (3.01, 3.02, 3.30 and 3.40) L1 data files. The V4 532nm daytime calibration coefficient and uncertainty anomalies are scaled to the means of the time series: $5.0619 \times 10^{10}$ $km^3 \cdot sr \cdot J^{-1} \cdot counts$ and $4.5088 \times 10^8$ $km^3 \cdot sr \cdot J^{-1} \cdot counts$ respectively.**

The impact of interpolating the high-latitude portions of the orbit where it is not possible to match daytime and nighttime attenuated scattering ratios (Sect. 3.6) can also be seen in the distribution of the 532 nm calibration coefficient uncertainties in Fig. 5(b). The saw tooth seasonal pattern of elevated uncertainty, greater than 1.5 times above the normalized mean, directly corresponds to those areas of interpolation. Though the time series of the uncertainty is fairly stable throughout the mission, there are pockets at the mid-latitudes (~1500 orbital elapsed seconds) in which there are localized spikes. These correspond to instances when there is a re-start of the calibration with a greater contribution of signals from the SAA. As previously discussed, the multi-averaging window technique mitigates the impact of the influence of signal variability in the SAA on the calibration through significant cross-track averaging. However, in the case of a calibration re-start the averaging window compresses and the impact of the SAA on the calibration uncertainties is amplified, though the overall mean is not.

The ratio between the V4 and V3 532 nm daytime calibration coefficients is shown in Fig. 5(c). In general, the mid-latitude differences, corresponding to 1200 – 1800 granule elapsed seconds, show differences in the range of ±5 %, with a global mean of 0.937 (6.3 %). This decrease in the calibration coefficient is expected, as there is an overall reduction in the 532 nm nighttime calibration coefficients of approximately the same magnitude (Kar et al., 2018). The high-latitude reduction of the calibration ratio to 0.95 and less corresponds closely with the uncertainty in Fig. 5(b). This reduction is also expected, because both the calibration and calibration uncertainty are interpolated to fit the neighboring night-time granules.





### 4.2 Zonal distributions of day and night attenuated scattering ratios

The performance of the new calibration algorithm can be evaluated by comparing zonal distributions of the day and night mean clear air attenuated scattering ratios in different altitude regimes. The calibration transfer region is the first altitude regime to be given attention. Given that the daytime calibration is scaled to the night-time, one should expect

to see that the daytime and nighttime attenuated scattering ratios should tightly follow each other within this altitude band. Figure 6 confirms this expectation. The red and blue curves show, respectively, mean daytime and nighttime calibration coefficients as a function of latitude, with the shaded areas around each curve delineating ±1 standard error about the mean. Comparisons are shown for each of the four seasons (January, April, July and October 2010). The SAA is excluded to minimize radiation-induced noise, and the L2 layer detection results are used to guarantee that

only clear-air regions are included.

From 60° north to south, for all four months, the mean nighttime attenuated scattering ratios fall consistently within the uncertainty of the corresponding daytime values. For data poleward of ± 60°, the seasonal impact of high-latitude interpolation due to non-coincident day-to-night matching is observed (Sect. 3.6). Below ~65° S, the attenuated scattering ratios for the daytime in January 2010 and the nighttime in July 2010 are distinctly elevated above the

general latitudinal trends. These could be caused by the presence of undetected polar stratospheric features in the July nighttime data, and by unusually high noise levels introduced by the southern auroral radiation belt (Hunt et al., 2009) in the January daytime data.

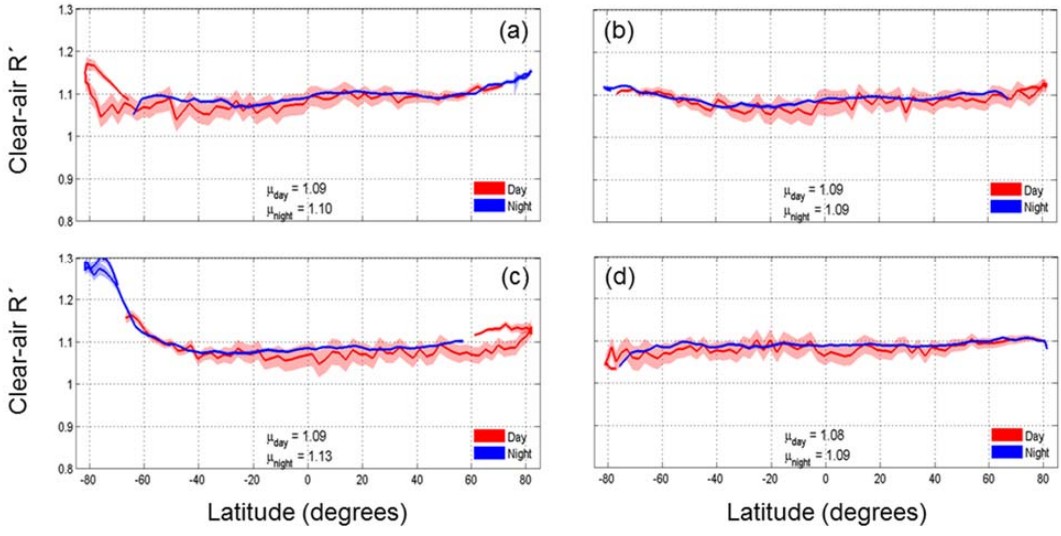

**Figure 6: Zonal clear-air attenuated scattering ratio (R´); means (solid lines) ± one standard error (shaded regions) for day and night in the calibration transfer regions for (a) January, (b) April, (c) July, and (d) October 2010. Global monthly means are given for both daytime ($\mu_{day}$) and nighttime ($\mu_{night}$).**



The altitude region between 24 and 30 km is also examined. These altitudes lie just above the top of the calibration transfer region used in the 532 nm daytime calibration, but below the 36–39 km region used by the 532 nm night-time calibration procedure. Thus, data within this region have not been used in any of the calibration procedures. Figure 7 shows the daytime–to–nighttime ratios of the clear air attenuated scattering ratios measured in the 24–30 km region

for both V3 (Fig. 7(a)) and V4 (Fig. 7(b)). The same months and data filtering procedures used in creating Fig. 6 are also used to construct Fig. 7. The V3 day–to–night ratios reveal high daytime biases of up to 20% in the mid-latitudes and 25% in the high-latitudes, with values above 1 consistently between ~50°S and ~60°N. The V4 day–to–night ratios eliminate the seasonal and latitudinal differences seen in V3. The V4 data are considerably more uniform and stable than the V3 data, with mean values of approximately one for all months and at all latitudes, confirming the

ability of the V4 calibration procedure to fully compensate for the high solar background noise levels and thermal beam steering effects that are constantly present in CALIOP's daytime measurements.

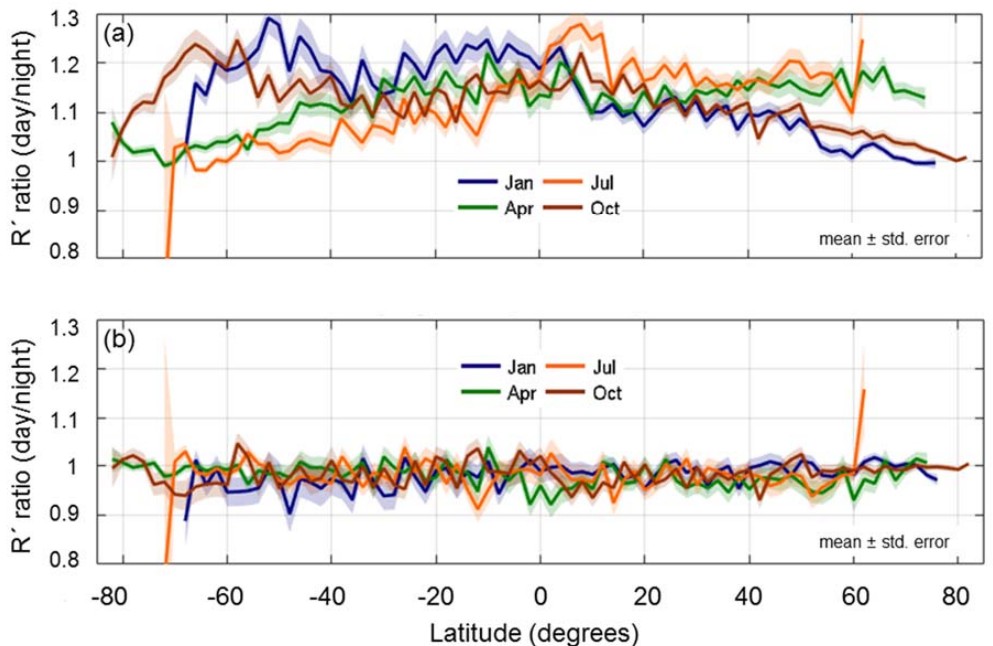

**Figure 7: Day / night ratio of clear-air attenuated scattering ratio (R´) mean ± one standard error at 24-30 km for (a) V3**
**and (b) V4 for January, April, July, and October 2010. The SAA has been removed.**

**4.3 Probability of feature detection using 1064 nm**

The V4 daytime calibration algorithm scans the uncalibrated 1064 nm measurements to ensure the presence of clear air down to the base of the calibration transfer regions. While the 532 nm channel is much more sensitive to the smaller aerosol particles that we expect to encounter most often in the stratosphere, the daytime calibration procedure



does not require that we identify pristine air parcels. Instead, we need only identify and remove relatively robust, spatially varying, and temporally transient features – i.e., those layers that are not expected to persist uniformly across extended day-night cycles – and for this task the 1064 nm detection capabilities should be sufficient.

To establish the performance capabilities of our 1064 nm feature detection approach, one year of 532 nm daytime
calibrations were regenerated using the more robust feature detection and clearing provided by the 532 nm detection methods of the L2 algorithm. In creating this second set of calibration coefficients, the V4 5 km merged layer product was used to identify those V4 calibration transfer regions where layers of any type are reported in the L2 data products, and regions identified as being feature-contaminated were excluded from the subsequent calculations. Like the L1 detection scheme, the L2 algorithm uses fixed frames of data, but with a maximum of 80 km rather than the 200 km
horizontal averages used in L1. The L2 technique also employs multi-pass averaging (5, 20 and 80km) and clearing to remove features detected at higher spatial resolutions prior to re-averaging and searching for features at coarser resolutions. The L2 532 nm algorithm implementation is thus capable of identifying features at much finer spatial scales than the 1064 nm version of the search routine implemented in L1.

Figure 8 shows the ratios of these two sets of 532 nm calibration coefficients for the entirety of 2015, plotted as a
function of latitude. While some latitudinal deviation is seen, the mean value (i.e., the black dashed line) varies by no more than ±0.5 % about the expected value of 1, indicating that feature clearing using the 1064 nm data introduces essentially negligible perturbations to the derived calibration coefficients. The solid blue line in Fig. 8 shows the ratios formed when the V3 calibration coefficients are divided by the regenerated V4 calibration coefficients. These values are comparable to those presented in Fig. 5(c) for the full mission V4/V3 comparison, and provide further
evidence that the 1064 nm detection technique, if not ideal, is nevertheless both robust and reliable.

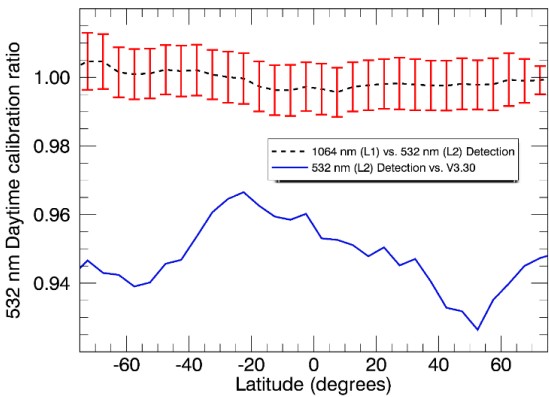

**Figure 8: Ratio of the V4 532 nm daytime calibrations derived based on 1064 nm detection technique (new L1 algorithm) to the calibrations derived by applying 532 nm feature clearing for 2015 in black. Red error-bars indicate that the mean calibration uncertainty. Blue line is the ratio of V4 to V3 daytime calibrations.**

Figure 9 provides a more detailed examination of the feature averaging and detection characteristics of the layers identified by the L2 532 nm feature detection algorithm and used in the re-calibration effort described in the previous paragraph. The plot is segregated by the effectiveness of the 1064 nm technique to detect features relative to the





averaging required (i.e., 5 km, 20 km, or 80 km) to detect layers when using the 532 nm L2 detection scheme. The distributions of detected and undetected L2 features are plotted as a function of layer integrated volume depolarization ($\delta_v$, x-axis) and 1064 nm integrated attenuated backscatter ($\gamma'_{1064}$, y-axis). Figure 9 indicates that detection failures by the 1064 nm technique are likely due to the insensitivity of the 1064 nm signal to smaller particles. Those layers that

are missed by the L1 1064 method, yet are detected by the L2 532 nm algorithm, most often have small IAB and low depolarization. It is also likely that the missed layers are being washed out at the 200 km 1064 nm detection resolution, and it takes a smaller spatial averaging window to isolate these feature within the averaged signal profiles.

The preponderance of the 1064 nm detection failures is seen in the bottom left corners of the plots in the right hand column of Fig. 9. These features, for which both $\gamma'_{1064}$ and $\delta_v$ are very low, are likely to be faint and perhaps even

persistent aerosol layers that consist of small particles that are not readily detectable at 1064 nm. For other features having low values of $\gamma'_{1064}$, the likelihood that the L2 algorithm has identified a false positive increases sharply as $\delta_v$ rises above ~0.7. While depolarization ratios approaching ~0.7 have been reported for contrail cirrus (e.g., Sassen and Hsueh, 1998) and have been occasionally observed in nighttime observations of polar stratospheric clouds (PSCs) (Pitts et al., 2009), values of this magnitude are highly unlikely for the bulk of the features that form in the stratospheric

regions searched during the 532 nm daytime calibration procedure. Furthermore, 98 % of all layers in this study having $\gamma'_{1064} < 0.005$ sr$^{-1}$ and $\delta_v > 0.7$ were detected during the daytime when PSCs are not present and when the general susceptibility of the signal to noise at high altitudes is at its maximum. In both cases, these missed features (or false positives) are not being removed and are included in the 532 nm daytime calibration calculations. But as noted earlier, the overall impact of including these features is negligible.



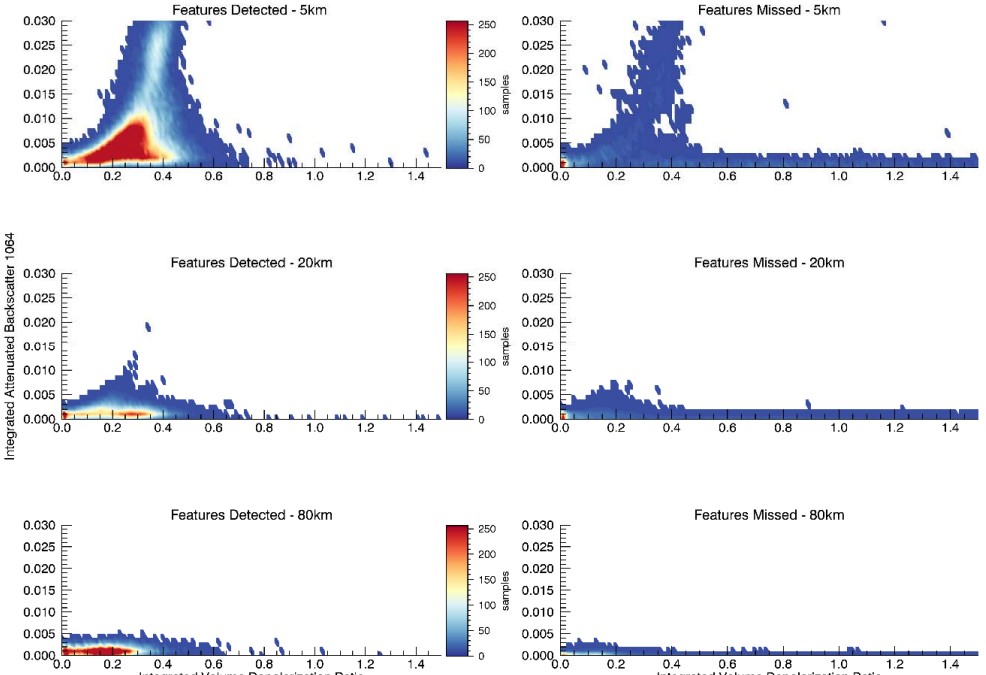

**Figure 9: Distribution of layer integrated volume depolarization ratio and 1064 nm integrated attenuated backscatter for all features contained in the transfer regions used for calibration. The left column are those instances when both the 532 nm and 1064 nm techniques have identified the prescience of a feature in the transfer regions, while the right column are those instances when 532 nm found a layer while the 1064 nm did not. Distribution is also broken by the horizontal averaging used by the 532 nm V4 L2 feature detection, 5km in the top row, 20km in the middle row and 80km in the bottom row.**

Figure 10 provides an example wherein the 1064 nm feature detection algorithm fails to identify legitimate features, and thus illustrates those circumstances in which calibration accuracy can be degraded by high biases in the nighttime attenuated scattering ratios. The orbit track begins just south and west of Africa, transiting over the southern oceans and extending into Antarctica at ~75° S. The scene is dominated by a mix of stratospheric and tropospheric clouds, with a single PSC of widely varying backscatter intensity continuously covering over half the along-track distance. Figure 10(d) compares L2 532 nm layer detections with the L1 1064 nm results. The tops of the layers identified by the V4 L2 layer detection scheme are shown by solid black lines, while the tops of layers detected by the L1 1064 nm method are shown in red diamonds. Also show are the base altitudes of the calibration transfer region (green lines) and the 532 nm mean attenuated scattering ratios (blue diamonds) computed within those calibration transfer regions where no layer was detected by the L1 1064 nm algorithm.

The tops of the tropospheric clouds north of 57° S are readily identified by the 1064 nm feature detection algorithm, but these lie below the base of the calibration transfer region. However, south of ~57° S an extended PSC, with top altitudes at roughly 25 km, lies well within the calibration transfer region. The boxed area in Fig. 10(d) (dashed red



lines) encloses a region where the 1064 nm feature detection algorithm consistently failed to detect layers that are reported in the V4 VFM. When comparing the 1064 nm feature detection results to the 532 nm total attenuated backscatter (Fig. 10(a)) and 1064 nm attenuated backscatter (Fig. 10(b)), it is clear that 1064 nm feature detection is successful for strongly scattered features, but may have difficulty identifying the weakly scattering features. The 532

nm mean attenuated scattering ratios within the calibration transfer regions where layers were not detected by the 1064 nm feature detection algorithm are, on average, 50% greater in this example than those in the neighboring clear air regions (i.e., ~1.5 vs. ~1.0 in the clear air regions). However, their ultimate impact on 532 nm daytime calibration is typically quite small due to extensive along-track and across-track averaging, as demonstrated in Fig. 8. And in this particular example, missed detections poleward of 67.5° S do not contribute to calibration biases because the

calibration coefficients at these latitudes are derived via interpolation, as described in Sect. 3.7.

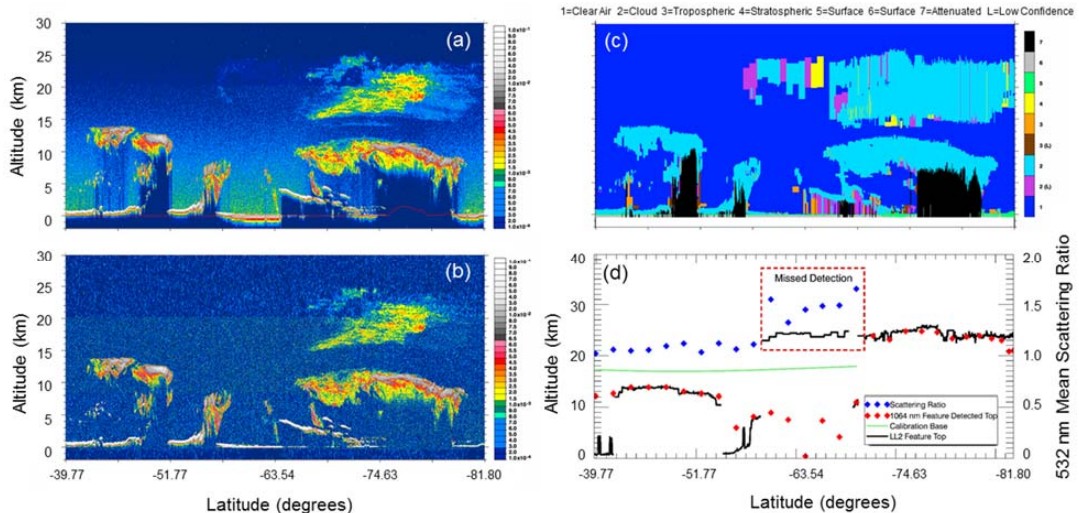

**Figure 10:** (a) 532 nm total attenuated backscatter; (b) 1064 nm attenuated backscatter; (c) vertical feature mask; and (d) layer detection results for July 15ᵗʰ, 2010 from 00:45Z to 59:07Z. In panel (d), the uppermost layer top altitudes (km)

detected by the 532 nm L2 algorithm are shown by black lines; layer top altitudes (km) detected by the 1064 nm L1 algorithm are shown by red diamonds; the base of the V4 calibration transfer region is shown by a green line; and blue diamonds show the 532 nm mean attenuated scattering ratios (unitless; right y-axis) in the calibration transfer regions where no layer was detected at 1064 nm.

### 4.4 Comparisons to HSRL measurements

From the beginning of the CALIPSO mission, the high spectral resolution lidar (HSRL) group at NASA-LaRC has acquired an extensive series of coincident airborne validation measurements. Following the release of the V3 L1 dataset in April 2010, Rogers et al. (2011) conducted an in-depth analysis comparing HSRL 532 nm attenuated backscatter coefficients measured along the CALIPSO orbit track to the 532 nm attenuated backscatter coefficients reported in the V3 CALIOP L1 data products. A major finding of this work showed that the CALIOP V3 daytime

attenuated backscatter data was biased low with respect to the coincident HSRL data by 2.9% ± 3.9%.



To characterize biases in the new V4 data set we replicated the Rogers study using a slightly larger coincident data set that includes additional overflights conducted since the original investigation. These include flights over the Caribbean (19 August 2010 – 28 September 2010), the DEVOTE field campaign (04 October 2011 – 08 October 2011), and flights over the Azores (17 October 2012) and Bermuda (10 June 2014 – 19 June 2014). In the process of reproducing the Rogers et al. (2011) V3 results, a bug was discovered in the code used to estimate the overlying two-way transmittance differences between the two sets of measurements (see Appendix A in Kar et al, 2018). Accounting for this error led to a small upward revision of the daytime biases in the V3 dataset, which we now estimate at 3.3% ± 3.1%. Running this same comparison using the CALIOP V4 data and the larger coincident HSRL–CALIOP data set shows that the bias between the two sets of daytime measurements has now decreased to 1.0% ± 3.5%. The differences between the revised V3 analyses and the new V4 analyses are illustrated in Fig. 11. Further reduction of the CALIOP–HSRL bias in future analyses is unlikely. In doing the comparisons, the HSRL signals are corrected for known attenuations that occur between the CALIPSO satellite altitude and the HSRL aircraft altitude (e.g., molecular and ozone attenuation). However, as explained in Kar et al. (2018),which focused on the nighttime comparisons between CALIOP and HSRL, the HSRL measurements cannot be corrected for any attenuation due to undetected cloud or aerosol layers in this altitude regime (e.g., the background stratospheric aerosol layer). Failure to correct for an undetected optical depth of 0.005 yields an attenuation bias of ~1%, a value that is essentially identical to our current estimate of the bias between CALIOP and HSRL.

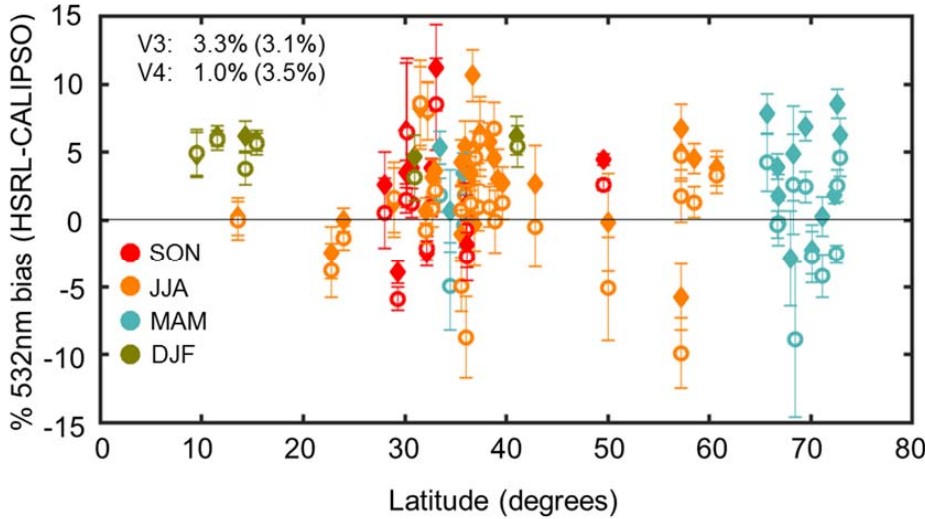

**Figure 11: Bias of the daytime 532 nm attenuated backscatter measured between HSRL and CALIPSO for several over-flight campaigns between 2006 and 2014 broken by season and latitudes. The comparisons used both V3 (solid diamonds) and V4 (open circles) L1 data. Each point represents the mean and uncertainty of the HSRL-CALIPSO difference for each of the 62 flights conducted.**




## 5 Concluding remarks

In this paper we have described the new procedures implemented in CALIOP's version 4 (V4) data release to better calibrate the 532 nm daytime measurements. Compared to version 3 (V3), the V4 updates deliver marked improvements in calibration accuracy and provide more realistic and comprehensive estimates of calibration

uncertainties. The new V4 algorithm keeps the underlying approach that was used in V3, wherein the 532 nm daytime calibration coefficients are scaled relative to the 532 nm nighttime coefficients, which are calculated using the highly reliable high altitude normalization technique. The simplified V4 calibration architecture reduces software coupling and increases cohesion by eliminating the need for multi-pass product generation cycle, which in turn enables a more direct computation of the calibration coefficients and their uncertainties. The V4 calibration performance meets pre-

defined expectations established from internal science impact testing, and fully satisfies numerous day-night consistency metrics. Elevating the calibration transfer region, coupled with a revised feature detection scheme that uses the uncalibrated 1064 nm measurement, has greatly increased the probability that the attenuated scattering ratios used in deriving the calibration coefficients are computed within clear air regions and largely eliminated the diurnal aerosol loading artifacts seen in V3. Independent validation using collocated high spectral resolution lidar

measurements shows a demonstrable improvement between CALIOP V3 and V4 daytime calibration, with the mean daytime bias between the two sets of measurements being reduced from 3% to approximately 1%.

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
