# Peer review of "CALIPSO Lidar Calibration at 532 nm: Version 4 Daytime Algorithm"

_Atmospheric Measurement Techniques, 2018_

## Referee Comment (RC1) · J. Yorks (Referee) · 11 Sep 2018

This paper provides an overview of CALIOP version 4 daytime 532 nm calibration algorithm and verification of the new algorithm. The quality of the daytime CALIOP data is very critical to data users of both level 1 and level 2 CALIPSO data products, and important to informing future missions. The algorithm is robust and I believe the CALIPSO team has done a great job improving the accuracy of the daytime level 1 data. The paper is well written, clear, and provides impressive results to a very challenging problem. It deserves to be published after a few minor revisions that I believe will strengthen the paper.

The 8 minor issues to be addressed are:

[Figure]

1) Diurnal variability of aerosols: On page 2 line 15, you discuss the assumption that the aerosols within the calibration transfer region are diurnally invariant. This is again discussed on page 5 lines 18-21. Please provide evidence that this is true or cite a paper that makes this claim to support this assumption.

2) Limitations of V3: On page 2 line 21, you say "First, the altitude of the V3 calibration transfer region was too low, and hence the assumed." Was a paper published already that shows this? If so, please cite it here.

3) Latency: Somewhere in Sections 2 and 3, please discuss the total latency of the 532 nm daytime calibration (V3 and V4). On page 3 lines 10-14 you discuss the need to use previous nighttime granules and on page 4 lines 14-15 you say clear air scattering ratios are accumulated for 7 days. Was the latency in V3 7 days? Did the changes for version 4 add additional time?

4) Polar Clouds: On page 8, lines 6-27 you discuss the new calibration transfer region. This is a good discussion, but I found myself wondering what impact PSC's have in the Polar region. Later, I read nice discussions on this in Sections 4.2 and 4.3. I suggest adding a sentence on page 8 stating that PSCs can introduce some uncertainty to the V4 532 nm daytime calibration constants and more details are discussed in those later sections.

5) Grammar errors: On page 8, line 15 the phrase "The work product from this study" is a bit confusing to me. I believe you mean that "The final result of this study...". On line 25 of that same page, you have two commas in a row. On page 23, line 13 there does not appear to be a space between "(2018)," and "which".

6) 1064 nm feature detection: I think this is a really good idea. One concern I have though is that you are only identifying layers that are > 1 km thick. Certainly, you can get some very thin volcanic or smoke plumes in the UTLS (I've seen them in CALIOP and CATS data). How much do you think these types of layers contribute to what you see in Figure 9? Please add a sentence or 2 on this subject to Section 4.3.

7) Interpolation of missing data: As I read Section 3.7, I found myself wondering how this interpolation may reduce the calibration accuracy. I later found a nice discussion on page 16. Please add a sentence to Section 3.7 that says something like "the implications of this interpolation on the accuracy of the calibration constant is discussed in Section 4.1 and Figure 5".

8) Figure 9: Something that is confusing me about this figure: Is the white color where the frequency equals 0 and blue is non-zero (1, 2, etc.). Or is the blue zero? The color bar would suggest the latter. Please try to clarify this.

––––––––––––––––––––––––––––––––

---

## Referee Comment (RC2) · Z. Wang (Referee) · 5 Oct 2018

The paper details the CALIPSO V4 daytime 532 nm calibration approach, which is critical for using CALIOP data. The paper is well organized, and the general approach is sound. But some details could be better presented. I suggest it for publication after the following comments are properly addressed.

1. Even the paper presented detail error estimations. But not all potential sources are included. Based on Figures 1,2, and 4, there are large calibration variations with time or locution. Although the potential mechanisms to day and night time calibration differences were discussed, what control these spatial variations during daytime were not touched. These daytime in-granule variations could indicate that there is a possibility

for large between granule variations, which could be a large random error source. Is there any way to quantify this?

2. Section 4.2 and Fig. 6: There are few major questions related to the discussion here. First, I don't think that the comparison gives you a real independent evaluation of daytime calibrations because your approach assumes that the day and night are same. The results only indicate that the approach is properly implemented. It is not clear which zonal clear air data are used here, all clear air or only in the calibration transferring zone? If the results are for the calibration transferring zone, the attenuated scattering ratio given in the figure is too high for me because the upper troposphere and low stratosphere have very low scattering ratio, especially under background conditions.

3. Page 4, line 9: should Eq. 2 be Eq. 3?

4. Page 5, Eq. (5): Is Cnight a constant here?

5. Figure1: use large font sizes for labels and legends.

6. Page 6, section 3.1: the baseline slope correction is hard to follow. Can you provide equations to support the discussion?

7. Page 8, lines 30-33: Why not using the new data to re-calculate altitude?

8. Page 10, line 24: Is 15 m here right?

9. Figure 3: Using a nighttime case to illustrate the approach is fine, but it will be good to see a daytime case because it is the focus of the paper. Due to the lower SNR, daytime data are challenging to handle.

---

## Author Comment (AC1) · 5 Nov 2018

**Responses to Reviewer #1 – John Yorks (amt-2018-206-RC1.pdf)**

1. Diurnal variability of aerosols: On page 2 line 15, you discuss the assumption that the aerosols within the calibration transfer region are diurnally invariant. This is again discussed on page 5 lines 18-21. Please provide evidence that this is true or cite a paper that makes this claim to support this assumption.

   This is discussed more in detail in section 3.2. In this section we make the argument that a region which is atmospherically stable and is decoupled (as much as one can) from the lower atmosphere would not have a difference in the aerosol loading between day and night. As noted in that section, using an isentropic surface to isolate atmospherically stable regions is fundamentally a more correct approach than using tropopause heights (i.e., as we did in V3). Hoskins (1991) develops a concept called the 'Overworld', which is a region defined by isentropic surfaces that do not cross the tropopause. This increased stability acts to cap motions from the lower troposphere, with the exceptions of forced events (strong convection, forced lifting, volcanic events, etc…). This paper is cited on line 20 of page 8.

   To further confirm our assertion, we discussed the diurnal variability of aerosols with Dr. Larry Thomason, who has expert knowledge of the spatial and temporal distributions of stratospheric aerosols derived over many decades from the analysis of data from multiple sensors. Dr. Thomason confirms that over his career he has never observed a diurnal variation in background stratospheric aerosol measurements that was not ultimately traced to some kind of instrument artifact. In fact, in a recently published review paper (Thomason et al. 2018), diurnal variation of stratospheric aerosol loading was not mentioned because (a) background stratospheric aerosols are not significantly photochemically active and (b) diurnal changes have been not observed in the historical measurement record (Larry Thomason, personal communication).

2. Limitations of V3: On page 2 line 21, you say "First, the altitude of the V3 calibration transfer region was too low, and hence the assumed." Was a paper published already that shows this? If so, please cite it here.

   This is discussed in more detail in Section 2 (Version 3 532 nm daytime calibration), in which to compensate for the inherent differences between day and night we had to introduce a scaling to correct the scattering ratios at the lower altitude.

   *Additional filtering and smoothing are applied to mitigate outliers. In particular, a minimum nighttime scattering ratio of 1.03 is used to compensate for diurnal differences in aerosol loading in the troposphere. Selection of this offset was based on observational analysis during development of the V3 algorithm, where it was noted that zonal distributions of attenuated scattering ratios in the calibration transfer regions fell below 1.0 in the tropics.*

   There was no V3 532 nm daytime algorithm paper per se, but Kathy Powell did describe this in detail in her ILRC papers and presentations from 2008 (new citation) and 2010 (already cited elsewhere). The addition of Kathy's 2008 paper is noted below (changes in red).

   *Calibration algorithms used in the version 3 (V3) series of L1 data products (Vaughan et al., 2018), released beginning in June 2009 are described in Hostetler et al., 2005, Powell et al., 2008, Powell et al., 2009, Powell et al., 2010, and Vaughan et al., 2010. Over the intervening years since the release of V3, several shortcomings have been identified in the 532 nm daytime calibration algorithm. First, the altitude of the V3 calibration transfer region was too low, and hence the assumed….*

   To further clarify this the proposed change to text (in red) is below, in which we also cite Kathy's ILRC presentation and ask the reader to look at Section 2.

*First, the altitude of the V3 calibration transfer region was too low, and hence the assumed diurnal invariance for the 532 nm daytime calibration was often not satisfied, as noted in Powell et al., 2010 and described further in Section 2.*

3. Latency: Somewhere in Sections 2 and 3, please discuss the total latency of the 532 nm daytime calibration (V3 and V4). On page 3 lines 10-14 you discuss the need to use previous nighttime granules and on page 4 lines 14-15 you say clear air scattering ratios are accumulated for 7 days. Was the latency in V3 7 days? Did the changes for version 4 add additional time?

Version 3:

Seven days specifies the ideal time period over which the V3 algorithm accumulates all of the scattering ratios needed to derive the V3 calibration correction factors. However, 7 days is not the total latency of this product. The 7-day averaging window works like a FIFO queue; as new data is accumulated, the oldest data is discarded, and the averages are recomputed. Our standard V3 data products are generated within 2 to 4 days from downlink, partially due to this calibration approach and also because we require a number of inputs that are not available at the time of down-link.

Our expedited V3 products are generated with 24-36 hours from downlink. We use a less robust multi-day mean calibration, as well as estimates on some of the other required information (e.g., estimated attitude and ephemeris). This are more tailored for near-real-time processing applications and not for rigorous science – which is noted on the CALIPSO website.

The proposed change in the text below, found on page 3, lines 12 – 14, explains that the mean correction factors are built from a sequence of correction factors created over several days, and that more specificity (the question you brought up) will be provided later in section 2.

*For any daytime granule, the 532 nm daytime calibration coefficients were then computed as the product of  a mean correction factor built from an accumulation of several days' worth of scattering ratios (discussed further in this section) and the mean 532 nm calibration coefficient from the previous nighttime granule (Powell et al., 2010).*

Version 4:

Like version 3, the version 4 approach also accumulates the scattering ratios over some fixed period of time. But, instead of using the previous 7 days, the V4 approach accumulates data over a fixed number of orbits. The number of orbits required is different for day and night, but can be denoted generically as 'N'. Calibration estimates are derived using data acquired over $\pm\frac{1}{2}N$ orbits about the current orbit. This strategy automatically introduces a 3-4 day latency in the ability to create the daytime L1 product. However, because V4 uses MERRA2 reanalysis data rather than the GMAO-FPIT (Forward Processing for Instrument Teams) data used in V3, the actual CALIOP data product latency is on the order of 6 to 10 weeks from downlink. GMAO typically delivers the MERRA2 product one month at a time, usually the middle of the next month. So, for instance, CALIPSO would get October 2017 no earlier than November 10-20[th], 2017.

Section 4, page 9 lines 12-15 (below) was updated to clarify that there was not a minimum window needed as in V3.

*Applying standard propagation of errors techniques to the daytime calibration equations shows that an averaging period of 105 consecutive orbits (i.e. over 7 days), centered on the orbit to be calibrated, should be sufficient to derive calibration coefficients with acceptably low random uncertainties. Unlike V3, which requires a minimum of 4 days to accumulate the required scattering ratios to build the mean correction factor, the V4 approach does*

*not have a set minimum number of orbits. Any reduction in the number of orbits used to generate the calibration coefficients will be reflected in the associated uncertainties.*

In addition to the changes cited above, an entirely new section (Section 3.9) was added to more completely describe the latencies of both the V3 (standard and expedited) and V4 (standard) products.

**3.9 Data latencies**

*Data latencies – i.e., the times between data acquisition and data product delivery – have also changed between V3 and V4. The CALIOP V3 standard data products are generated within 3 to 5 days from downlink, partially due to the V3 calibration approach but also because the analyses require a number of ancillary inputs that are not immediately available (Winker et al., 2009). The latency for the V4 standard products is considerably longer. Because V4 uses the MERRA-2 meteorological data rather than the GMAO FPIT products, V4 standard products are typically not available until 6 to 10 weeks from downlink. However, the V3 expected products continue to be available with 24 – 36 hours from data acquisition (i.e., ~12 hours from data downlink). The expedited processing uses a faster (albeit less robust) calibration strategy, as well as estimates for some of the other required information (e.g., platform attitude and ephemeris). The expedited products are tailored specifically for near-real-time processing applications, whereas the standard products are designed for rigorous scientific analyses.*

4. Polar Clouds: On page 8, lines 6-27 you discuss the new calibration transfer region. This is a good discussion, but I found myself wondering what impact PSC's have in the Polar region. Later, I read nice discussions on this in Sections 4.2 and 4.3. I suggest adding a sentence on page 8 stating that PSCs can introduce some uncertainty to the V4 532 nm daytime calibration constants and more details are discussed in those later sections.

We have added the following sentence (in red) to the paragraph on page 8, lines 23-27.

*Two additional safeguards are used to avoid possible contamination of the clear-air attenuated scattering ratios. First, to both guard against features intruding into the lower stratosphere, and because the algorithm uses a climatological monthly mean 400K surface as the lower limit, , an additional altitude offset of 2 km is applied to further elevate the base of the calibration transfer region. Secondly, since the stratosphere is not entirely devoid of features, the algorithm employs a 1064 nm feature detection technique, as discussed in Sect. 3.4, to exclude cloud and aerosol layers from the calibration averaging scheme. In particular, the presence of undetected polar stratospheric clouds (PSCs) in the calibration transfer regions can introduce high biases into the calibration coefficient estimates. The potential impacts of feature contamination of the calibration transfer regions are discussed in detail in Sections 4.2 and 4.3.*

5. Grammar errors: On page 8, line 15 the phrase "The work product from this study" is a bit confusing to me. I believe you mean that "The final result of this study…" On line 25 of that same page, you have two commas in a row. On page 23, line 13 there does not appear to be a space between "(2018)," and "which".

We have changed page 8 line 15 to change the phrase as you suggested.

* The final result of this study is a comprehensive set of lookup tables derived from 5 years of V3 L2 5 km cloud profile data and the corresponding GEOS 5 FP-IT (Forward Processing for Instrument Teams) Version 5.91…*

The grammatical issues noted for *(1) line 25 of page* 8 and *(2) line 13 on page 23* have been corrected.

6.  1064 nm feature detection: I think this is a really good idea. One concern I have though is that you are only identifying layers that are > 1 km thick. Certainly, you can get some very thin volcanic or smoke plumes in the UTLS (I've seen them in CALIOP and CATS data). How much do you think these types of layers contribute to what you see in Figure 9? Please add a sentence or 2 on this subject to Section 4.3.

    Consider the right column of Figure 9, in which the 1064 nm failed to identify a layer in the UTLS but the 532 nm did. Of the layers we failed to detect at 1064 nm, only relatively small fractions have vertical extents of 1 km or less. Segregated according to the horizontal averaging required for layer detection, these fractions are 13.3%, 21%, and 12% at averaging resolutions of, respectively, 5 km, 20 km, and 80 km. Figure xx (below) shows 1064 nm IAB (y-axis) versus 532 nm depolarization ratio (x-axis) for these 'less than 1 km' layers, scaled as in Figure 9.

[Figure]

[Figure]

[Figure]

Figure XX: Identical approach used to generate Figure 9, but only using layers with a depth of less than 1 km

We have added the following sentence (in red) to the paragraph on page 20, lines 8-15.

*The preponderance of the 1064 nm detection failures is seen in the bottom left corners of the plots in the right hand column of Fig. 9. These features, for which both $\gamma'_{1064}$ and $\delta_v$ are very low, are likely to be faint and perhaps even persistent aerosol layers that consist of small particles that are not readily detectable at 1064 nm. For other features having low values of $\gamma'_{1064}$, the likelihood that the L2 algorithm has identified a false positive increases sharply as $\delta_v$ rises above ~0.7. That is not to say that all are false positives. As noted in Section 3.4, features are identified when the 1064 nm signal is above a threshold for 1 km or more. This will likely exclude the detection of thin features (< 1 km vertically) which may be strongly scattering but have small particle sizes (e.g., volcanic ash or elevated smoke). Also, while depolarization ratios approaching ~0.7 have been reported for contrail cirrus (e.g., Sassen and Hsueh, 1998) and have been occasionally observed in nighttime observations of polar stratospheric clouds (Pitts et al., 2009), values of this magnitude are highly unlikely for the bulk of the features that form in the stratospheric regions searched during the 532 nm daytime calibration procedure.*

7. Interpolation of missing data: As I read Section 3.7, I found myself wondering how this interpolation may reduce the calibration accuracy. I later found a nice discussion on page 16. Please add a sentence to Section 3.7 that says something like "the implications of this interpolation on the accuracy of the calibration constant is discussed in Section 4.1 and Figure 5".

We have added the following sentence (in red) to the paragraph on page 14, lines 8-17.

*Calibration coefficients derived from the day-to-night ratio of attenuated scattering ratios can only be calculated within those portions of an orbit in which both day and night observations are acquired. Because of the illumination patterns in the polar regions during the solstice seasons there are no matching daytime and nighttime samples near the poles in summer and winter. This seasonal high latitude lack of matching day and night data is accounted for by anchoring the ends of the neighboring daytime and nighttime calibration coefficient curves and interpolating between these end points. Where there are neither daytime and/or nighttime samples, the 532 nm daytime calibration coefficient and coefficient uncertainty curves are linearly interpolated as a function of orbital elapsed time anchored to the nearest neighboring 532 nm nighttime calibration. Figure 4 shows an example. In this orbit from July 2010, the day-to-night terminator occurs at ~60° N, and thus no corresponding nighttime measurements are available over the final ~1100 seconds of the daytime granule. The impact of this interpolation on the accuracy of the calibration coefficients and uncertainty estimates is discussed further in Section 4.1.*

8. Figure 9: Something that is confusing me about this figure: Is the white color where the frequency equals 0 and blue is non-zero (1, 2, etc.). Or is the blue zero? The color bar would suggest the latter. Please try to clarify this

Figure 9 uses IDL color table 72, which is Color Brewer Scheme Red-Yellow-Blue. White is where the frequency is 0, while the minimum of the blue distribution is 1. In order to better convey all intended information, the figure was recreated with a different color bar that goes from blue to yellow, a re-scaling of the x and y axis, a single color bar at the bottom of the image, and a change from sampling counting to sample frequency. Sample frequency, derived independently for each of the 6 plots, is computed by dividing samples by the maximum sample. These changes do not impact the discussion contained in the section related to this figure.

[Figure]

References:

Thomason, L. W., Ernest, N., Millán, L., Rieger, L., Bourassa, A., Vernier, J.-P., Manney, G., Luo, B., Arfeuille, F., and Peter, T.: A global space-based stratospheric aerosol climatology: 1979–2016, Earth Syst. Sci. Data, 10, 469-492, https://doi.org/10.5194/essd-10-469-2018, 2018.

---

## Author Comment (AC2) · 5 Nov 2018

**Responses to Reviewer #2 – Z. Wang (amt-2018-206-RC2.pdf)**

1. Even the paper presented detail error estimations. But not all potential sources are included. Based on Figures 1, 2, and 4, there are large calibration variations with time or locution. Although the potential mechanisms to day and night time calibration differences were discussed, what control these spatial variations during daytime were not touched. These daytime in-granule variations could indicate that there is a possibility for large between granule variations, which could be a large random error source. Is there any way to quantify this?

   We do not dig deeply into "what control(s) these spatial variations during daytime" because the full extent of all mechanisms involved is not precisely known. But we can make some definitive statements about the general nature of the underlying cause. Post launch thermal modeling by the CALIOP engineers at Ball Aerospace Technology Corporation demonstrates that the predominate source of these variations is thermally induced misalignment of the CALIOP transmitter and receiver (e.g., see Hunt et al., 2009; Powell et al., 2010; and Stephens et al., 2010).

   As illustrated in Figure 5, these thermal beam steering effects manifest themselves differently for the two lasers. Of more relevance, perhaps, is the fact that the relative magnitudes of these effects within any given orbit vary seasonally as a result of changes in solar incidence angle with respect to the satellite. Our averaging scheme is specifically designed to capture and characterize these changes. Furthermore, because the thermal mass of the instrument is large and essentially constant, any changes in the sunlight-induced, time-varying thermal stress profile from orbit to orbit are expected to be very small, and thus would not serve as "a large random error source" for "large between granule variations" in the 532 nm daytime calibration procedure.

2. Section 4.2 and Fig. 6: There are few major questions related to the discussion here. First, I don't think that the comparison gives you a real independent evaluation of daytime calibrations because your approach assumes that the day and night are same. The results only indicate that the approach is properly implemented.

   The only independent evaluation of the daytime calibration is done is section 4.4 (Comparisons to HSRL measurements), where the backscatters between the HSRL and CALIPSO are compared. It is not the intent of figure 6 to show independence, rather to verify that the scaling of the day to the night is working. This is noted on page 17, lines 4 – 6, where we say:

   *Given that the daytime calibration is scaled to the night-time, one should expect to see that the daytime and nighttime attenuated scattering ratios should tightly follow each other within this altitude band. Figure 6 confirms this expectation.*

   If the algorithm was not working properly we would not expect to see the very close correspondence that is shown in the plot.

   It is not clear which zonal clear air data are used here, all clear air or only in the calibration transferring zone? If the results are for the calibration transferring zone, the attenuated scattering ratio given in the figure is too high for me because the upper troposphere and low stratosphere have very low scattering ratio, especially under background conditions.

   Figure 6 is made from data in the calibration transfer region, as noted in the figure title. Figure 7 looks at a fixed altitude band above the calibration target region of 24-30km, but is a ratio between the day and the night, not the actual scattering ratios.

   The two figures below show the 532 nm mean clear-air attenuated scattering ratios measured by CALIPSO and reported in our Lidar Level 2 product as a function of height and altitude for the month of January 2010, which corresponds to Figure 6(a). These clear-air attenuated scattering ratios are computed using only those profiles in which are found to be "feature-free" by the CALIOP layer

detection algorithm. A mean scattering ratio of ~1.1, both in the calibration transfer region and again from 24–30 km, is wholly consistent with the zonal means shown in figure 6 below. Note that the daytime image shows enhanced residual effects of failed detections of (presumably) subvisible cirrus in the tropical tropopause layer (i.e., at ~17 km between -20°S and 10°N).

[Figure]

[Figure]

3. Page 4, line 9: should Eq. 2 be Eq. 3?

You are correct, I meant to reference equation 3. The correction to page 4 line 9 is high-lighted in red.

*…, are calculated using the same formula as in Eq. 3, except using nighttime signals…*

4. Page 5, Eq. (5): Is Cnight a constant here?

The parameter contained in Eq. (5) is defined in the text on page 4, lines 2 - 5:

*The clear-air attenuated scattering ratios in the V3 8-12 km calibration transfer region were assumed to be diurnally invariant. Based on this assumption, initial estimates of the mean attenuated scattering ratios, $\bar{R}'_{day,initial}$, are calculated for each daytime frame **using the mean of the 532 nm calibration coefficients, $\tilde{C}_{night}$, computed during the previous nighttime granule.***

(Bold gold emphasis added.) It is a constant in terms of Eq. (5).

5. Figure1: use large font sizes for labels and legends

The label font (Helvetica) and size (12 pt) is consistent with the other figures in the paper. The figure has been recreated to increase the size of the legend (Night and Day) to also be 12pt Helvetica.

[Figure]

6. Page 6, section 3.1: the baseline slope correction is hard to follow. Can you provide equations to support the discussion?

We have replaced the first two paragraphs of section 3.1 with a more in-depth explanation of the baseline slope corrections. In doing so, we have also replaced 'baseline slope' with 'baseline shape', as using the term 'slope' implies a linear correction, whereas in fact the correction we actually apply is a quadratic function of the measured background light intensity.

*The V4 calibration procedure applies two new corrections to the daytime signal prior to the calibration: an adjustment to remove photomultiplier (PMT) baseline shapes and an updated day-to-night gain ratio. The motivation and implementation of these two corrections are discussed in the paragraphs below.*

*The output of PMTs exposed to constant background light (e.g., sunlight reflected from dense water clouds) typically increases with time after the PMT is gated on, thus generating a signal-induced baseline shape that varies as a function of the background light level. Prior to launch, the baseline shapes for the CALIOP detectors were repeatedly measured in the laboratory, and the magnitudes of the required signal adjustments were found to be quite small relative to the atmospheric signals typically measured in the troposphere. Consequently, because the prelaunch daytime calibration strategy was simply to interpolate daytime calibration coefficients between neighboring nighttime molecular normalizations (Hostetler et al., 2006; Powell et al., 2008), baseline shape corrections were deemed to be unnecessary and thus were not implemented. This assessment changed with the V4 redesign of the daytime calibration algorithms. The V4 daytime calibration relies on highly averaged daytime measurements in the middle-to-lower stratosphere where the expected molecular signals are substantially weaker, and hence biases due to baseline shape artifacts are potentially significant. To mitigate*

*these concerns, we used prelaunch laboratory measurements together with post-launch extended background measurements acquired periodically throughout the mission to characterize the PMT baseline shapes:*

$$\text{shape}(z, B) = (z_{\text{offset}} - z)(X_1 B + X_2 B^2)10^{G/20}. \tag{6}$$

*This approximation is a function of both altitude (z) and background light intensity (B). $X_1$ and $X_2$ are polynomial coefficients separately determined for the 532 nm parallel and perpendicular channels, B is the measured background light level for each laser pulse, G is the channel-dependent electronic amplifier gain, z is the measurement altitude for each range bin in a CALIOP backscatter profile, and $z_{offset} = 72.5$ km is the midpoint of CALIOP's high-altitude digitizer DC offset measurement region (Hunt et al., 2009). Shape correction profiles are computed on a shot-by-shot basis and applied to all data acquired during daytime measurements.*

7. Page 8, lines 30-33: Why not using the new data to re-calculate altitude?

Page 8, lines 28-33:

*At the time of the V4 algorithm development and deployment, GMAO provided an updated meteorological reanalysis product, MERRA-2 (Modern-Era Retrospective analysis for Research and Applications, Version 2) (Gelaro, 2017), which includes Microwave Limb Sounder (MLS) temperatures and is a marked improvement over earlier GMAO-FPIT products. This new meteorological data was incorporated into the V4.10 L1 and L2 data products, but was not used to re-compute the 400 K altitudes used by the 532 nm daytime calibration algorithm to set the calibration transfer region base altitude.*

For a majority of the development of the new algorithm we only had the GMAO FP-IT data, which was used for V3. The selection of the MERRA-2 was made much later in the algorithm development cycle. The rationale for not using MERRA-2 to rebuild the 400K tables was two-fold. First, based on internal analysis of the differences, the 400K line did not deviate significantly between GMAO FP-IT and MERRA-2. Second, since we are adding a 2km correction above the 400K line we felt that that provided enough margin to account for any possible differences if they indeed existed.

8. Page 10, line 24: Is 15 m here right?

Page 10, lines 23-25:

*...where N represents the number of profiles averaged and $RMS_{1064}$ is the root-mean-square of the baseline signal measured on-board the satellite for each laser pulse at 15 m vertical resolution and subsequently recorded in the L1 data products. The layer detection threshold, $T(z)$, is then computed as a function of the on-board averaging using...*

Yes, 15 m is correct here. This is the fundamental resolution of the measurements collected, before the on-board averaging is carried out.

9. Figure 3: Using a nighttime case to illustrate the approach is fine, but it will be good to see a daytime case because it is the focus of the paper. Due to the lower SNR, daytime data are challenging to handle.

The 1064 nm detection is applied to both day and night orbits, so the purpose of figure was strictly illustrative and meant show a simplified case. I chose a night-time orbit because I wanted Figure 3(a), (c) – (e) to be relatively clean – precisely because of the lower SNR. As noted, the daytime 532 nm total attenuated backscatter case contains more noise (see figure below).

The other take-home from Figure 3 is that what is actually being used is the region above the 400K line (yellow line in figure 3(b)), and not what was done in V3 (red boxes in figure 3(b))).

References

Hunt, W. H., Winker, D. M., Vaughan, M. A., Powell, K. A., Lucker, P. L., and Weimer, C.: CALIPSO Lidar Description and Performance Assessment, J. Atmos. Oceanic Technol., 26, 1214–1228, doi:10.1175/2009JTECHA1223.1, 2009.

Powell, K. A., Hostetler, C. A., Liu, Z., Vaughan, M. A., Kuehn, R. E., Hunt, W. A., Lee, K. P., Trepte, C. R, Rogers, R. R, Young, S. A., and Winker, D. M.: CALIPSO Lidar Calibration Algorithms: Part I - Nighttime 532 nm Parallel Channel and 532 nm Perpendicular Channel, J. Atmos. Oceanic Technol., 26, 2015–2033, doi:10.1175/2009JTECHA1242.1, 2009.

Stephens, M., Weimer, C., Saiki, E., and Lieber, M.: On-orbit models of the CALIOP lidar for enabling future mission design, Proc. SPIE 7807, Earth Observing Systems XV, 78070F, doi:10.1117/12.860900